

# Chemical composition and the potential for proteomic transformation in cancer, hypoxia, and hyperosmotic stress

Jeffrey M. Dick

Wattanothaipayap School, Chiang Mai, Thailand

## ABSTRACT

The changes of protein expression that are monitored in proteomic experiments are a type of biological transformation that also involves changes in chemical composition. Accompanying the myriad molecular-level interactions that underlie any proteomic transformation, there is an overall thermodynamic potential that is sensitive to microenvironmental conditions, including local oxidation and hydration potential. Here, up- and down-expressed proteins identified in 71 comparative proteomics studies were analyzed using the average oxidation state of carbon ($Z_C$) and water demand per residue ($\overline{n}_{H_2O}$), calculated using elemental abundances and stoichiometric reactions to form proteins from basis species. Experimental lowering of oxygen availability (hypoxia) or water activity (hyperosmotic stress) generally results in decreased $Z_C$ or $\overline{n}_{H_2O}$ of up-expressed compared to down-expressed proteins. This correspondence of chemical composition with experimental conditions provides evidence for attraction of the proteomes to a low-energy state. An opposite compositional change, toward higher average oxidation or hydration state, is found for proteomic transformations in colorectal and pancreatic cancer, and in two experiments for adipose-derived stem cells. Calculations of chemical affinity were used to estimate the thermodynamic potentials for proteomic transformations as a function of fugacity of $O_2$ and activity of $H_2O$, which serve as scales of oxidation and hydration potential. Diagrams summarizing the relative potential for formation of up- and down-expressed proteins have predicted equipotential lines that cluster around particular values of oxygen fugacity and water activity for similar datasets. The changes in chemical composition of proteomes are likely linked with reactions among other cellular molecules. A redox balance calculation indicates that an increase in the lipid to protein ratio in cancer cells by 20% over hypoxic cells would generate a large enough electron sink for oxidation of the cancer proteomes. The datasets and computer code used here are made available in a new R package, **canprot**.

Corresponding author
Jeffrey M. Dick, j3ffdick@gmail.com

# INTRODUCTION

The relationship between cells and tissue microenvironments is a topic of vital importance for cancer biology. Because of rapid cellular proliferation and irregular vascularization, tumors often develop regions of hypoxia (*Höckel & Vaupel, 2001*). Tumor microenvironments also exhibit abnormal ranges of other physical-chemical variables, including hydration state (*McIntyre, 2006*; *Abramczyk et al., 2014*).

Some aspects of the complex metazoan response to hypoxia are mediated by hypoxia-inducible factor 1 (HIF-1). HIF-1 is a transcription factor that is tagged for degradation in normoxic conditions. Under hypoxia, the degradation of HIF-1 is suppressed; HIF-1 can then enter the nucleus and activate the transcription of downstream targets (*Semenza, 2003*). Indeed, transcriptional targets of HIF-1 are found to be differentially expressed in proteomic datasets for laboratory hypoxia (*Cifani et al., 2011*; *McMahon et al., 2012*). However, proteomic studies of cells in hypoxic conditions provide many examples of proteins that are not directly regulated by HIF-1 (*McMahon et al., 2012*; *Fuhrmann et al., 2013*), and cancer proteomic datasets also include many proteins that are not known to be regulated by HIF-1.

The complexity of the underlying regulatory mechanisms (*McMahon et al., 2012*) and the large differences between levels of gene expression and protein abundance (*van den Beucken et al., 2011*; *Cifani et al., 2011*; *Ho et al., 2016*) present many difficulties for a bottom-up understanding of global proteomic trends. As a counterpart to molecular explanations, a systems perspective can incorporate higher-level constraints (*Drack & Wolkenhauer, 2011*). A commonly used metaphor in systems biology is attractor landscapes. The basins of attraction are defined by dynamical systems behavior, but in many cases are analogous to minimum-energy states in thermodynamics (*Emmeche, Koppe & Stjernfelt, 2000*; *Enver et al., 2009*). Nevertheless, little attention has been given to the thermodynamic potential that is inherent to the compositional difference between the up-expressed and down-expressed proteins in proteomic experiments. Such a high-level perspective may require concepts and language that differ from those applicable to molecular interactions (*Ellis, 2015*).

To better understand the microenvironmental context for compositional changes, this study uses proteomic data as input into a descriptive thermodynamic model. First, a compositional analysis of differentially (up- and down-) expressed proteins identifies consistent trends in the oxidation and hydration states of proteomes of colorectal cancer (CRC), pancreatic cancer, and cells exposed to hypoxia or hyperosmotic stress. These results lay the groundwork for using a thermodynamic model to quantify environmental constraints on the potential for proteomic transformation. Finally, the Discussion section explores some implications of the hypothesis that elevated synthesis of lipids provides an electron sink for the oxidation of proteomes. In this situation, some cancer systems may develop an abnormally large redox disproportionation between pools of cellular biomacromolecules.

## METHODS

### Data sources

Tables 1–4 present the sources of data. Protein IDs and expression (up/down or abundance ratios) were found in the literature, often being reported in the supporting information (SI) or supplementary (suppl.) tables. In some cases, source tables were further processed, using fold-change and significance cutoffs that, where possible, are based on statements made in the primary publication. The data are stored as *.csv files in the R package **canprot**, which was developed during this study (see http://github.com/jedick/canprot) and is provided as Dataset S1.

**Table 1** **Selected proteomic datasets for colorectal cancer.**[*] Here and in Tables 2–4, $n_1$ and $n_2$ stand for the numbers of down- and up-expressed proteins, respectively, in each dataset.

| Set | $n_1$ | $n_2$ | Description | Set | $n_1$ | $n_2$ | Description |
|---|---|---|---|---|---|---|---|
| ⓐ | 57 | 70 | T/N | ⓢ | 73 | 175 | MSS-type T/N[a] |
| ⓑ | 101 | 28 | CRC C/A[a] | ⓣ | 79 | 677 | T/N |
| ⓒ | 87 | 81 | CIN C/A[a] | ⓤ | 55 | 68 | CM T/N[b] |
| ⓓ | 157 | 76 | MIN C/A[a] | ⓥ | 33 | 37 | stromal T/N[a] |
| ⓔ | 43 | 56 | biomarkers up/down | ⓦ | 51 | 55 | chromatin-binding C/A |
| ⓕ | 48 | 166 | stage I/normal[b] | ⓧ | 58 | 65 | epithelial A/N |
| ⓖ | 77 | 321 | stage II/normal[b] | ⓨ | 44 | 210 | tissue secretome T/N[a] |
| ⓗ | 61 | 57 | microdissected T/N[b] | ⓩ | 113 | 66 | membrane enriched T/N |
| ⓘ | 71 | 92 | adenoma/normal[a] | Ⓐ | 1061 | 1254 | A/N |
| ⓙ | 109 | 72 | stage I/normal[a] | Ⓑ | 772 | 1007 | C/A |
| ⓚ | 164 | 140 | stage II/normal[a] | Ⓒ | 879 | 1281 | C/N |
| ⓛ | 63 | 131 | stage III/normal[a] | Ⓓ | 123 | 75 | stromal AD/NC[a] |
| ⓜ | 42 | 26 | stage IV/normal[a] | Ⓔ | 125 | 60 | stromal CIS/NC[a] |
| ⓝ | 72 | 45 | T/N | Ⓕ | 99 | 75 | stromal ICC/NC[a] |
| ⓞ | 335 | 288 | A/N | Ⓖ | 191 | 178 | biopsy T/N[b] |
| ⓟ | 373 | 257 | C/A | Ⓗ | 113 | 86 | AD/NC[a] |
| ⓠ | 351 | 232 | C/N | Ⓘ | 169 | 138 | CIS/NC[a] |
| ⓡ | 75 | 61 | poor/good prognosis[b] | Ⓙ | 129 | 100 | ICC/NC[a] |

**Notes.**

Abbreviations: T, tumor; N, normal; C, carcinoma or adenocarcinoma; A, adenoma; CM, conditioned media; AD, adenomatous colon polyps; CIS, carcinoma *in situ*; ICC, invasive colonic carcinoma; NC, non-neoplastic colonic mucosa.

[*] ⓐ Source: Table 1 and Suppl. Data 1 of *Watanabe et al. (2008)*. ⓑ ⓒ ⓓ Nuclear matrix proteome; chromosomal instability (CIN), microsatellite instability (MIN), or both types (CRC). Source: Suppl. Tables 5–7 of *Albrethsen et al. (2010)*. ⓔ Candidate serum biomarkers. Source: Table 4 of *Jimenez et al. (2010)*. ⓕ ⓖ Source: Suppl. Table 4 of *Xie et al. (2010)*. ⓗ Source: Suppl. Table 4 of *Zhang et al. (2010)*. ⓘ ⓙ ⓚ ⓛ ⓜ Source: Suppl. Table 9 of *Besson et al. (2011)*. ⓝ Source: Suppl. Table 2 of *Jankova et al. (2011)*. ⓞ ⓟ ⓠ Source: Table S8 of *Mikula et al. (2011)*. ⓡ Source: extracted from Suppl. Table 5 of *Kim et al. (2012)*, including proteins with abundance ratio >2 or <0.5. ⓢ Microsatellite stable (MSS) type CRC tissue. Source: Suppl. Table 4 of *Kang et al. (2012)*. ⓣ Source: Suppl. Table 4 of *Wiśniewski et al. (2012)*. ⓤ Source: Suppl. Table 2 of *Yao et al. (2012)*. ⓥ Source: Table 1 of *Mu et al. (2013)*. ⓦ Source: Table 2 of *Knol et al. (2014)*. ⓧ Source: Table III of *Uzozie et al. (2014)*. ⓨ Source: Suppl. Table 1 of *de Wit et al. (2014)*. ⓩ Source: Supporting Table 2 of *Sethi et al. (2015)*. Ⓐ Ⓑ Ⓒ Source: SI Table 3 of *Wiśniewski et al. (2015)*. Ⓓ Ⓔ Ⓕ Source: Suppl. Table S3 of *Li et al. (2016)*. Ⓖ Source: extracted from SI Table S3 of *Liu et al. (2016)*, including proteins with *p*-value < 0.05. Ⓗ Ⓘ Ⓙ Source: Suppl. Table 4 of *Peng et al. (2016)*.
[a] Gene names or GI numbers were converted to UniProt IDs using the UniProt mapping tool.
[b] IPI numbers were converted to UniProt IDs using the DAVID conversion tool.

Sequence IDs were converted to UniProt IDs using the UniProt mapping tool (http://www.uniprot.org/mapping/) or the gene ID conversion tool of DAVID 6.7 (https://david.ncifcrf.gov/conversion.jsp). For proteins where the automatic conversions produced no matches, manual searches in UniProt were performed using the gene names or protein descriptions. If specified (i.e., as UniProt IDs with suffixes), particular isoforms of the proteins were used. Obsolete or secondary IDs reported for some proteins were updated to reflect current, primary IDs (`uniprot_updates.csv` in Dataset S1). Any duplicated IDs listed as having opposite expression ratios were excluded from the comparisons here.

Amino acid sequences of human proteins were taken from the UniProt human reference proteome. Sequences of proteins in other organisms and of human proteins not contained in the reference proteome were downloaded from UniProt or the NCBI website (for one study reporting GI numbers; see Table 4). Amino acid compositions were computed using

**Table 2   Selected proteomic datasets for pancreatic cancer.**[*]

| Set | $n_1$ | $n_2$ | Description | Set | $n_1$ | $n_2$ | Description |
|---|---|---|---|---|---|---|---|
| (a) | 41 | 69 | T/N | (l) | 29 | 73 | FFPE PC/AIP[c] |
| (b) | 60 | 88 | T/N[a] | (m) | 53 | 73 | FFPE PC/CP[c] |
| (c) | 48 | 54 | T/N[a] | (n) | 83 | 32 | low-grade T/N[a] |
| (d) | 19 | 95 | CP/N[a] | (o) | 224 | 176 | high-grade T/N[a] |
| (e) | 28 | 29 | T/N | (p) | 208 | 219 | T/N (no DM)[a] |
| (f) | 38 | 45 | T/N[b] | (q) | 56 | 167 | T/N (DM)[a] |
| (g) | 207 | 152 | FFPE T/N[a] | (r) | 227 | 148 | LCM PDAC/ANT[c] |
| (h) | 108 | 86 | accessible T/N[c] | (s) | 65 | 34 | T/N |
| (i) | 38 | 47 | FFPE T/N[c] | (t) | 35 | 51 | mouse 2.5 w T/N[a] |
| (j) | 78 | 57 | T/N[a] | (u) | 40 | 73 | mouse 3.5 w T/N[a] |
| (k) | 257 | 456 | T/N[a] | (v) | 49 | 84 | mouse 5 w T/N[a] |
|  |  |  |  | (w) | 37 | 108 | mouse 10 w T/N[a] |

**Notes.**

Abbreviations: T, tumor; N, normal; CP, chronic pancreatitis; AIP, autoimmune pancreatitis; PC, pancreatic cancer; DM, diabetes mellitus; PDAC, pancreatic ductal adenocarcinoma; ANT, adjacent normal tissue; FFPE, formalin-fixed paraffin-embedded; LCM, laser-capture microdissection; NP, normal pancreas.

[*](a) Pooled tissue samples of PC and matched normal tissue from 12 patients. Source: Tables 2 and 3 of *Lu et al. (2004)*. (b) Two PC and two NP samples. Source: Tables 1 and 2 of *Chen et al. (2005)*. (c) Large-scale immunoblotting (PowerBlot) of 8 tissue specimens of pancreatic intraepithelial neoplasia compared to NP and CP. Source: Table 2 of *Crnogorac-Jurcevic et al. (2005)*. (d) Tissue specimens from patients with CP and 10 control specimens from patients with NP. Source: Table 1 of *Chen et al. (2007)*. (e) 12 carcinoma samples (PDAC), 12 benign pancreatic cystadenomas and 10 normal tissues adjacent to the PDAC primary mass. Source: Table 1 of *Cui et al. (2009)*. (f) Source: extracted from Table S2 of *McKinney et al. (2011)*. (g) PDAC compared to NP. Source: Suppl. Table 3 of *Pan et al. (2011)*. (h) Potentially accessible proteins in fresh samples of PC tumors (three patients) vs normal tissue (two patients with NP and one with CP). Source: extracted from the SI Table of *Turtoi et al. (2011)*. (i) 11 tissue specimens containing >50% cancer and 8 unmatched, uninvolved tissues adjacent to pancreatitis. Source: Suppl. Tables 2 and 3 of *Kojima et al. (2012)*. (j) Fresh-frozen PDAC tissue specimens from seven patients vs a pooled mixture of three normal main pancreatic duct tissue samples. Source: extracted from SI Table S3 of *Kawahara et al. (2013)*, including proteins with an expression ratio >2 [or <0.5] in at least five of the seven experiments and ratio >1 [or <1] in all experiments. (k) Frozen samples of PDAC tumors vs adjacent benign tissue from four patients. Source: Suppl. Table 2 of *Kosanam et al. (2013)*. (l) (m) Tissue samples from three patients with PC vs 3 patients with AIP or three patients with CP. Source: extracted from Tables 2, 3, and 4 of *Paulo et al. (2013)*. (n) (o) 12 samples each (pooled) of low-grade tumor or high-grade tumor vs non-tumor. Source: extracted from Suppl. Tables S4 and S5 of *Wang et al. (2013b)*, including proteins with ratios ≥3/2 or ≤2/3 for at least two of the four groups, and with expression differences for all four groups in the same direction. (p) (q) Source: extracted from Suppl. Tables S3 and S4 of *Wang et al. (2013a)*, including proteins with >3/2 or <2/3 fold change in at least 3 of 4 iTRAQ experiments for different pooled samples. (r) LCM of CD24[+] cells from PDAC vs CD24[−] cells from adjacent normal tissue (ANT). Source: SI Table S5 of *Zhu et al. (2013)*. (s) Matched PDAC and normal tissue from nine patients. Source: extracted from SI Table S5 of *Iuga et al. (2014)*, excluding "not passed" proteins (those with inconsistent regulation). (t) (u) (v) (w) PDAC tumors in transgenic mice vs pancreas in normal mice, at time points of 2.5, 3.5, 5 and 10 weeks. Source: Suppl. Table of *Kuo et al. (2016)*.

[a]Gene names, IPI numbers or UniProt names were converted to UniProt IDs using the UniProt mapping tool.
[b]IPI numbers were converted to UniProt IDs using the DAVID conversion tool.
[c]Includes differentially expressed proteins shared between groups and proteins identified in only one group.

functions in the **CHNOSZ** package (*Dick, 2008*) or the ProtParam tool on the UniProt website. The amino acid compositions are stored in *.Rdata files in Dataset S1.

R (*R Core Team, 2016*) and R packages **canprot** (this study) and **CHNOSZ** (*Dick, 2008*) were used to process the data and generate the figures with code specifically written for this study, which is provided in Dataset S2.

**Table 3  Selected proteomic datasets for hypoxia and reoxygenation experiments or growth in 3D culture.**[*]

| Set | $n_1$ | $n_2$ | Description | Set | $n_1$ | $n_2$ | Description | Set | $n_1$ | $n_2$ | Description |
|---|---|---|---|---|---|---|---|---|---|---|---|
| ⓐ | 37 | 24 | U937[a] | ⓚ | 56 | 40 | THP-1 | ⓥ | 113 | 154 | CRC-derived SPH |
| ⓑ | 41 | 22 | placental secretome | ⓛ | 178 | 77 | A431 Hx48 | ⓦ | 127 | 292 | HepG2/C3A SPH |
| ⓒ | 71 | 19 | B104 | ⓜ | 69 | 54 | A431 Hx72 | ⓧ | 53 | 72 | HeLa |
| ⓓ | 87 | 28 | DU145[a] | ⓝ | 48 | 36 | A431 ReOx | ⓨ | 137 | 64 | U87MG and 786-O |
| ⓔ | 29 | 21 | SK-N-BE(2)c; IMR-32 | ⓞ | 141 | 64 | SH-SY5Y | ⓩ | 129 | 141 | HCT116 transcription[a] |
| ⓕ | 53 | 65 | H9C2[b] | ⓟ | 65 | 34 | A431 Hx48-S | Ⓐ | 469 | 1024 | HCT116 translation[a] |
| ⓖ | 409 | 337 | MCF-7 SPH P5 | ⓠ | 137 | 61 | A431 Hx72-S | Ⓑ | 66 | 50 | adipose-derived SC[a] |
| ⓗ | 248 | 214 | MCF-7 SPH P2 | ⓡ | 56 | 49 | A431 ReOx-S | Ⓒ | 65 | 27 | cardiomyocytes CoCl$_2$[a] |
| ⓘ | 48 | 52 | SPH perinecrotic[a] | ⓢ | 74 | 44 | A431 Hx48-P | Ⓓ | 35 | 69 | cardiomyocytes SAL[a] |
| ⓙ | 101 | 186 | SPH necrotic[a] | ⓣ | 67 | 53 | A431 Hx72-P | Ⓔ | 116 | 225 | HT29 SPH |
| | | | | ⓤ | 41 | 31 | A431 ReOx-P | | | | |

**Notes.**

Abbreviations: U937, acute promonocytic leukemic cells; B104, rat neuroblastoma cells; DU145, prostate carcinoma cells; SK-N-BE(2)c; IMR-32; SH-SY5Y, neuroblastoma cells; H9C2, rat heart myoblast; MCF-7, breast cancer cells; THP-1, macrophages; A431, epithelial carcinoma cells; Hx48, hypoxia 48 h; Hx72, hypoxia 72 h; ReOx, hypoxia 48 h followed by reoxygenation for 24 h; -S, supernatant fraction; -P, pellet fraction; SPH, spheroids; HepG2/C3A, hepatocellular carcinoma cells; U87MG, glioblastoma; 786-O, renal clear cell carcinoma cells; HCT116; HT29, colon cancer cells; SC, stem cells; SAL, salidroside.

[*]ⓐ 2% O$_2$ vs normoxic conditions. Source: Table 1 of *Han et al. (2006)*. ⓑ 1% vs 6% O$_2$. Source: Tables 2 and 3 of *Blankley et al. (2010)*. ⓒ Expression ratios HYP/LSC (oxygen deprivation/low serum control) >1.2 or <0.83. Source: calculated using data from Suppl. Table 2 of *Datta et al. (2010)*, including proteins with *p*-value < 0.05 and EF < 1.4. ⓓ Translationally regulated genes. Source: Suppl. Tables 1–4 of *van den Beucken et al. (2011)*. ⓔ 1% O$_2$ for 72 h vs standard conditions. Source: Suppl. Table 1(a) of *Cifani et al. (2011)*. ⓕ Hypoxic vs control conditions for 16 h. Source: Suppl. Table S5 of *Li et al. (2012)*. ⓖ ⓗ Tumorspheres (50 to 200 µm diameter) at passage 5 (P5) or 2 (P2) compared to adherent cells. Source: Sheets 2 and 3 in Table S1 of *Morrison et al. (2012)*. ⓘ ⓙ Perinecrotic and necrotic regions compared to surface of multicell spheroids (∼600 µm diameter) (expression ratios <0.77 or >1.3). Source: Suppl. Table 1C of *McMahon et al. (2012)*. ⓚ Incubation for several days under hypoxia (1% O$_2$). Source: Suppl. Table 2A of *Fuhrmann et al. (2013)* (control virus cells). ⓛ ⓜ ⓝ Source: extracted from Suppl. Table 1 of *Ren et al. (2013)*, including proteins with iTRAQ ratios <0.83 or >1.2 and *p*-value < 0.05. ⓞ 5% O$_2$ vs atmospheric levels of O$_2$ (normalized expression ratio >1.2 or <0.83). Source: SI table of *Villeneuve et al. (2013)*. ⓟ ⓠ ⓡ ⓢ ⓣ ⓤ The comparisons here include proteins with *p* < 0.05. Source: Suppl. Table S1 of *Dutta et al. (2014)*. ⓥ Organotypic spheroids (∼250 µm diameter) vs lysed CRC tissue. Source: extracted from Table S2 of *Rajcevic et al. (2014)*, filtered as follows: at least two of three experiments have differences in spectral counts, absolute overall fold change is at least 1.5, and *p*-value is less than 0.05. ⓦ SPH vs classical cell culture (2D growth) (log$_2$ fold change at least ±1). Source: P1_Data sheet in the SI of *Wrzesinski et al. (2014)*. ⓧ 1% vs 19% O$_2$. Source: Table S1 of *Bousquet et al. (2015)*. ⓨ 1% O$_2$ for 24 h (fold change <0.5 or >1 for proteins detected in only hypoxic or only normoxic conditions). Source: Table S1 of *Ho et al. (2016)*. ⓩ Ⓐ Microarray analysis of differential gene expression in the transcriptome (total rRNA) and translatome (polysomal/total RNA ratio) of cells grown in normal and hypoxic (1% O$_2$) conditions. Source: data file supplied by Ming-Chih Lai (*Lai, Chang & Sun, 2016*). Ⓑ ASC from three donors cultured for 24 h in hypoxic (1% O$_2$) vs normoxic (20% O$_2$) conditions. Source: Tables 1 and 2 of *Riis et al. (2016)*. Ⓒ Ⓓ Rat cardiomyocytes treated with CoCl$_2$ (hypoxia mimetic) vs control or with SAL (anti-hypoxic) vs CoCl$_2$. Source: SI Tables 1S and 2S of *Xu et al. (2016)*. Ⓔ 800 µm spheroids vs 2D monolayers. Source: Tables S1a–b of *Yue et al. (2016)*.

[a]Gene names, GI numbers, or other IDs were converted to UniProt IDs using the UniProt mapping tool.

[b]IPI numbers were converted to UniProt IDs using the DAVID conversion tool.

## Measures of compositional oxidation and hydration state

Two compositional metrics that afford a quantitative description of proteomic data, the average oxidation state of carbon ($Z_C$) and the water demand per residue ($\overline{n}_{H_2O}$), are briefly described here.

The oxidation state of atoms in molecules quantifies the degree of electron redistribution due to bonding; a higher oxidation state signifies a lower degree of reduction. Although calculations of oxidation state from molecular formulas necessarily make simplifying assumptions regarding the internal electronic structure of molecules, such calculations may be used to quantify the flow of electrons in chemical reactions, and the oxidation state concept is useful for studying the transformations of complex mixtures of organic molecules. For example, calculations of the average oxidation state of carbon provide insight on the processes affecting the decomposition of carbohydrate, protein and lipid fractions of natural organic matter (*Baldock et al., 2004*). Moreover, oxidation state can be regarded as an ensemble property of organic systems (*Kroll et al., 2015*). See *Dick (2016)*

**Table 4  Selected proteomic datasets for hyperosmotic stress experiments.**[*]

| Set | $n_1$ | $n_2$ | Description | Set | $n_1$ | $n_2$ | Description |
|---|---|---|---|---|---|---|---|
| ⓐ | 38 | 44 | *S. cerevisiae* VHG 2 h[a] | ⓝ | 49 | 28 | eel gill[a] |
| ⓑ | 33 | 62 | *S. cerevisiae* VHG 10 h[a] | ⓞ | 78 | 77 | *S. cerevisiae* t30a[b] |
| ⓒ | 18 | 65 | *S. cerevisiae* VHG 12 h[a] | ⓟ | 67 | 67 | *S. cerevisiae* t30b[b] |
| ⓓ | 63 | 94 | mouse pancreatic islets | ⓠ | 87 | 87 | *S. cerevisiae* t30c[b] |
| ⓔ | 148 | 44 | adipose-derived stem cells | ⓡ | 25 | 38 | IOBA-NHC |
| ⓕ | 17 | 11 | ARPE-19 25 mM | ⓢ | 105 | 96 | CAUCR succinate tr.[a] |
| ⓖ | 21 | 24 | ARPE-19 100 mM | ⓣ | 209 | 142 | CAUCR NaCl tr.[a] |
| ⓗ | 114 | 61 | ECO57 25 °C, $a_w$ 0.985[a] | ⓤ | 33 | 33 | CAUCR succinate pr.[a] |
| ⓘ | 238 | 61 | ECO57 14 °C, $a_w$ 0.985[a] | ⓥ | 33 | 27 | CAUCR NaCl pr.[a] |
| ⓙ | 263 | 56 | ECO57 25 °C, $a_w$ 0.967[a] | ⓦ | 294 | 205 | CHO all[a] |
| ⓚ | 372 | 73 | ECO57 14 °C, $a_w$ 0.967[a] | ⓧ | 66 | 75 | CHO high[a] |
| ⓛ | 32 | 39 | Chang liver cells 25 mM | ⓨ | 14 | 28 | *Yarrowia lipolytica*[b] |
| ⓜ | 19 | 50 | Chang liver cells 100 mM | ⓩ | 160 | 141 | *Paracoccidioides lutzii* [a] |

**Notes.**

Abbreviations: VHG, very high glucose; ARPE-19, human retinal pigmented epithelium cells; ECO57, *Escherichia coli* O157:H7 Sakai; IOBA-NHC, human conjunctival epithelial cells; CAUCR, *Caulobacter crescentus*; tr, transcriptome; pr, proteome; CHO, Chinese hamster ovary cells.

[*] ⓐⓑⓒ VHG (300 g/L) vs control (20 g/L). The comparisons here use proteins with expression ratios <0.9 or >1.1 and with *p*-values < 0.05. Source: SI Table of *Pham & Wright (2008)*. ⓓ 24 h at 16.7 mM vs 5.6 mM glucose. Source: extracted from Suppl. Table ST4 of *Waanders et al. (2009)*; including the red- and blue-highlighted rows in the source table (those with ANOVA *p*-value < 0.01), and applying the authors' criterion that proteins be identified by 2 or more unique peptides in at least 4 of the 8 most intense LC-MS/MS runs. ⓔ 300 mOsm (control) or 400 mOsm (NaCl treatment). Source: Suppl. Table 1 of *Oswald et al. (2011)*. ⓕ ⓖ Mannitol-balanced 5.5 (control), 25 or 100 mM D-glucose media. Source: Table 1 of *Chen et al. (2012)*. ⓗ ⓘ ⓙ ⓚ Temperature and NaCl treatment (control: 35 °C, $a_w$ 0.993). Source: Suppl. Tables S13–S16 of *Kocharunchitt et al. (2012)*. ⓛ ⓜ 5.5 (control), 25 or 100 mM D-glucose. Source: Table 1 of *Chen et al. (2013)*. ⓝ Gill proteome of Japanese eel (*Anguilla japonica*) adapted to seawater or freshwater. Source: protein IDs from Suppl. Table 3 and gene names of human orthologs from Suppl. File 4 of *Tse et al. (2013)*. ⓞ ⓟ ⓠ Multiple experiments for 30 min after transfer from YPKG (0.5% glucose) to YNB (2% glucose) media. Source: extracted from Suppl. Files 3 and 5 of *Giardina, Stanley & Chiang (2014)*, using the authors' criterion of *p*-value < 0.05. ⓡ 280 (control), 380, or 480 mOsm (NaCl treatment) for 24 h. Source: Table 2 of *Chen et al. (2015)*. ⓢ ⓣ ⓤ ⓥ Overnight treatment with a final concentration of 40/50 mM NaCl or 200 mM sucrose vs M2 minimal salts medium plus glucose (control). Source: Table S2 of *Kohler et al. (2015)*. ⓦ ⓧ 15 g/L vs 5 g/L (control) glucose at days 0, 3, 6, and 9. The comparisons here use all proteins reported to have expression patterns in Cluster 1 (up) or Cluster 5 (down), or only the proteins with high expression differences (ratio ≤ −0.2 or ≥0.2) at all time points. Source: SI Table S4 of *Liu et al. (2015)*. ⓨ 4.21 osmol/kg vs 3.17 osmol/kg osmotic pressure (NaCl treatment). Source: Table 1 of *Yang et al. (2015)*. ⓩ 0.1 M KCl (treatment) vs medium with no added KCl (control). Source: Suppl. Tables 2 and 3 of *da Silva Rodrigues et al. (2016)*.

[a] Gene names, GI numbers, or NCBI RefSeq accessions were converted to UniProt IDs using the UniProt mapping tool.
[b] Amino acid sequences were obtained for the listed GI numbers using Batch Entrez (https://www.ncbi.nlm.nih.gov/sites/batchentrez).

for additional references where organic and biochemical reactions have been characterized using the average oxidation state of carbon.

Despite the large size of proteins, their relatively simple primary structure means that $Z_C$ can be computed using the elemental abundances in any particular amino acid sequence (*Dick, 2014*):

$$Z_C = \frac{-h + 3n + 2o + 2s + z}{c}. \tag{1}$$

In this equation, $c$, $h$, $n$, $o$, and $s$ are the elemental abundances in the chemical formula $C_c H_h N_n O_o S_s^z$ for a specific protein with total charge $z$. Note, however, that ionization by gain or loss of protons alters charge and the number of H equally, so has no effect on the

value of $Z_C$; for ease of computation, $Z_C$ is calculated here for proteins in their completely non-ionized forms.

In contrast to the elemental stoichiometry in Eq. (1), a calculation of the hydration state must account for the gain or loss of $H_2O$. In the biochemical literature, "protein hydration" or water of hydration refers to the effective (time-averaged) number of water molecules that interact with a protein (*Timasheff, 2002*). These dynamically interacting molecules form a hydration shell that has important implications for crystallography and enzymatic function, but hydration numbers have been measured for few proteins and are difficult to compute, especially for the many proteins with unknown tertiary structure. Thus, the structural hydration of proteins identified in proteomic datasets generally remains unquantified.

A different concept of hydration state arises by considering the chemical components that make up proteins. A componential analysis is a method of projecting the composition of a molecule using specified chemical formula units as the components, or basis species. The notion of components is central to chemical thermodynamics (*Gibbs, 1875*); the choice of components determines the thermodynamic variables (chemical potentials), and a careful choice leads to more convenient representations of the compositional and energetic constraints on reactions (e.g. *Zhu & Anderson, 2002*).

The components, or basis species, consist of a minimum number of species whose compositions can be linearly combined to represent the composition of any protein. The 20 proteinogenic amino acids are together composed of five elements (C, H, N, O, S), so five basis species are needed to represent the primary sequences of proteins. As noted previously (see references in *Dick, 2016*), all possible combinations of basis species lead to thermodynamically consistent models, but are differently suited to making interpretations. *Dick (2016)* proposed using $C_5H_{10}N_2O_3$, $C_5H_9NO_4$, $C_3H_7NO_2S$, $O_2$, and $H_2O$ as a basis for assessing compositional differences in proteomes. The first three formulas correspond to glutamine (Q), glutamic acid (E), and cysteine (C).

To account for protein ionization, a proton can be included in the basis, which is now referred to as "QEC+". Using the QEC+ basis, the stoichiometric projection of a protein with formula $C_cH_{h+z}N_nO_oS_s^z$, where $z$ is the charge of the protein and $h$ is the number of H in the fully nonionized protein, is represented by

$$n_{Cys}C_3H_7NO_2S + n_{Glu}C_5H_9NO_4 + n_{Gln}C_5H_{10}N_2O_3$$
$$+ n_{H_2O}H_2O + n_{O_2}O_2 + zH^+ \rightarrow C_cH_{h+z}N_nO_oS_s^z. \tag{R1}$$

To compare the compositions of different-sized proteins, the stoichiometric coefficients in Reaction (R1) can be divided by the sequence length (number of amino acids) of the protein. The length-normalized coefficients, written with an overbar, include the per-residue water demand for formation of a protein ($\overline{n}_{H_2O}$). This componential "hydration state" is used in this study, and should not be confused with the structural biochemical "protein hydration" mentioned above.

The primary reason for choosing the QEC+ basis instead of others lies in the relation of the compositional variables representing oxidation and hydration state ($\overline{n}_{O_2}$ and

$\overline{n}_{H_2O}$) with each other and with $Z_C$. It is important to note that $Z_C$ is a measure of oxidation state that is independent of the choice of basis species. Smoothed scatter plots of $\overline{n}_{H_2O}$ vs $Z_C$ and $\overline{n}_{O_2}$ vs $Z_C$ are shown in Fig. S1 for the 21,006 human proteins in the UniProt reference proteome. The plots in the top row of this figure are made using the QEC basis (which is equivalent to the QEC+ basis for the plotted variables) while those in the bottom row are made using the basis species $CO_2$, $NH_3$, $H_2S$, $H_2O$, and $O_2$; these inorganic species are often used to balance reactions in geochemical models. It is apparent from Fig. S1 that, using the QEC basis, $\overline{n}_{O_2}$ is highly positively correlated with $Z_C$, and $\overline{n}_{H_2O}$ shows a slight negative correlation with $Z_C$. Accordingly, in the QEC basis, $\overline{n}_{O_2}$ is a strong indicator of oxidation state, while $\overline{n}_{H_2O}$ represents a distinct compositional variable. In contrast, the plots in the bottom row of Fig. S1 show a moderate positive correlation between $\overline{n}_{O_2}$ and $Z_C$ and a stronger negative correlation between $\overline{n}_{H_2O}$ and $Z_C$. Using that basis would therefore weaken the interpretation of $\overline{n}_{O_2}$ as an indicator of oxidation state and of $\overline{n}_{H_2O}$ as a distinct compositional variable. The relations among $\overline{n}_{H_2O}$, $\overline{n}_{O_2}$, and $Z_C$ also vary between basis species consisting of different combinations of amino acids; those differences together with biological considerations support the choice of QEC instead of other amino acids (*Dick, 2016*).

In summary, Reaction (R1) is not a mechanism for protein synthesis, but is a projection of any protein's elemental composition into chemical components, i.e., the basis. Compared to a basis composed of simpler inorganic species, the QEC+ basis reduces the projected codependence of oxidation and hydration state in proteins, unfolding a compositional dimension that can enrich a thermodynamic model.

## RESULTS

### Colorectal cancer

The progression of colorectal cancer (CRC) begins with the formation of numerous non-cancerous lesions (adenoma), which may remain undetectable. Over time, a small fraction of adenomas develop into malignant tumors (carcinoma) (*Jimenez et al., 2010*; *Wiśniewski et al., 2015*). Publicly available datasets reporting a minimum of ca. 30 up- and 30 down-expressed proteins for tissue samples of CRC, and one meta-analysis of serum biomarkers, were compiled recently (*Dick, 2016*). These same datasets are listed in Table 1, with one newer addition (dataset ◎; *Liu et al., 2016*).

Many aspects of the experimental methods, statistical tests, and bioinformatics analyses used to identify significantly up-expressed and down-expressed proteins vary considerably among studies. The comparisons here are made without any control of this variability. Although particular comparisons may reflect study-specific conditions and methods, visualization of the chemical compositions of proteins for many datasets can reveal general features of the cancer phenotype.

For each dataset, Table 1 lists the numbers of down-expressed ($n_1$) and up-expressed ($n_2$) proteins in cancer relative to normal tissue. For datasets comparing different stages of cancer progression, groups $n_1$ and $n_2$ correspond to the down- and up-expressed proteins in the more advanced stage (e.g., carcinoma) compared to the less advanced stage (e.g.,
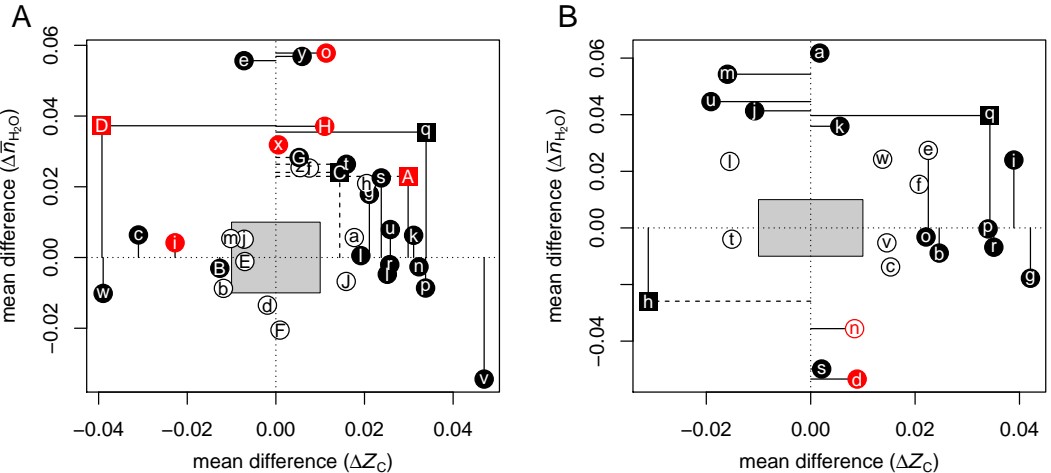

**Figure 1** **Compositional analysis of differential protein expression in (A) colorectal cancer and (B) pancreatic cancer.** The plots show differences ($\Delta$) between the mean for up-expressed and the mean for down-expressed proteins of average oxidation state of carbon ($Z_C$) and water demand per residue ($\bar{n}_{H_2O}$) for each dataset from Tables 1 and 2. Red colors highlight (A) adenoma/normal comparisons or (B) chronic pancreatitis/normal or low-grade tumor/normal comparisons. Here and in Fig. 2, filled points and dashed lines indicate $p < 0.05$; solid lines are drawn instead if the common language effect size is $\geq 60\%$ or $\leq 40\%$.

adenoma). Mean values of average oxidation state of carbon ($Z_C$; Eq. (1)) and water demand per residue ($\bar{n}_{H_2O}$; Reaction (R1)) were calculated for the up- and down-expressed groups of proteins, together with the corresponding mean differences ($\Delta Z_C$ and $\Delta \bar{n}_{H_2O}$ for the means of up- minus down-expressed groups), $p$-values, and effect sizes. These values are listed in Table S1. Figure S2 shows the mean values of $Z_C$ and $\bar{n}_{H_2O}$ for the up- and down-expressed proteins together in a single plot (lettered point symbols for down-expressed and arrowheads for up-expressed proteins). Because of the high variability of mean values among datasets, compositional trends between up- and down-expressed proteins are difficult to interpret using Fig. S2. Therefore, the differences in mean values between up- and down-expressed proteins ($\Delta Z_C$ and $\Delta \bar{n}_{H_2O}$) are plotted in this paper.

Figure 1A shows $\Delta \bar{n}_{H_2O}$ vs $\Delta Z_C$ for the CRC datasets. The gray boxes cover the range from $-0.01$ to $0.01$ for each of the variables. To draw attention to the largest and most significant changes, filled points and dashed lines indicate mean differences with a $p$-value (Wilcoxon test) less than 0.05; solid lines indicate mean differences with a common language effect size (CLES) $\geq 60\%$ or $\leq 40\%$. The common language statistic "is the probability that a score sampled at random from one distribution will be greater than a score sampled from some other distribution" (McGraw & Wong, 1992). Here, CLES is calculated as the percentage of pairings of individual proteins with a positive difference in $Z_C$ or $\bar{n}_{H_2O}$ between the up- and down-expressed groups from all possible pairings between the groups. Point symbols are squares if the $p$-values for both $Z_C$ and $\bar{n}_{H_2O}$ are less than 0.05, or circles otherwise.

The plot illustrates that proteins up-expressed in carcinoma relative to normal tissue most often have significantly higher $Z_C$ [ⓖ ⓚ ⓛ ⓝ ⓟ ⓡ ⓢ ⓤ ⓥ ①], $\bar{n}_{H_2O}$ [ⓒ ⓞ ⓣ ⓧ ⓨ ⓓ Ⓖ

ⓗ], or both [ⓖ Ⓐ ©] (see also *Dick, 2016*). The red points in the plot highlight the datasets for adenoma/normal comparisons [ⓘ ⓞ ⓧ Ⓐ Ⓓ ⓗ]. Most of these exhibit a significant positive $\Delta \overline{n}_{H_2O}$ but not the large increase in $Z_C$ found for many of the carcinoma/normal comparisons.

## Pancreatic cancer

Many proteomic studies have been performed to investigate the differences between normal pancreas (NP) and pancreatic adenocarcinoma (PDAC). Proteomic studies also address the inflammatory conditions of autoimmune pancreatitis, which is sometimes misidentified as carcinoma (*Paulo et al., 2013*), and chronic pancreatitis, which is associated with increased cancer risk (*Chen et al., 2007*). Searches for proteomic data were aided by the reviews of *Pan et al. (2013)* and *Ansari et al. (2014)*. Table 2 lists selected datasets reporting at least ca. 25 up-expressed and 25 down-expressed proteins.

The compositional comparisons in Fig. 1B show that up-expressed proteins in pancreatic cancer often have significantly higher $Z_C$ [ⓑ © ⓖ ⓘ ⓞ ⓟ ⓠ ⓡ]. A dataset obtained for pancreatic cancer associated with diabetes mellitus (*Wang et al., 2013a*) [ⓖ] has both significantly higher $Z_C$ and $\overline{n}_{H_2O}$. Only one dataset, from a study that targeted accessible proteins (*Turtoi et al., 2011*) [ⓑ], is characterized by a large negative mean difference of $\Delta Z_C$. Some other datasets that do not have significantly different $Z_C$ exhibit higher $\overline{n}_{H_2O}$ in cancer compared to non-cancerous (normal or pancreatitis) tissue [ⓐ ⓙ ⓚ ⓜ ⓤ]. Two of the four datasets with negative $\Delta \overline{n}_{H_2O}$ [ⓓ ⓗ ⓝ ⓢ] were obtained from studies of chronic pancreatitis (*Chen et al., 2007*) or low-grade tumors (*Wang et al., 2013b*) (red points in Fig. 1B); another used a procedure to isolate accessible proteins (*Turtoi et al., 2011*) [ⓑ], while the remaining low-$\Delta \overline{n}_{H_2O}$ dataset [ⓢ] may be an outlier in terms of mean chemical composition (Fig. S2). Therefore, the datasets with positive $\Delta \overline{n}_{H_2O}$ and/or $\Delta Z_C$ likely reflect a general characteristic of pancreatic cancer.

## Hypoxia and 3D culture

Hypoxia refers to oxygen concentrations that are lower than normal physiological levels. Hypoxia is a factor in many pathological conditions, including altitude sickness, stroke, and cardiac ischemia (e.g., *Datta et al., 2010*; *Li et al., 2012*; *Fuhrmann et al., 2013*). In tumors, irregular vascularization and abnormal perfusion contribute to the formation of hypoxic regions (*Höckel & Vaupel, 2001*). A related situation is the growth in the laboratory of 3D cell cultures (e.g., tumor spheroids), instead of two-dimensional growth on a surface. In 2D monolayers, all cells are exposed to the gas phase, but interior regions of 3D cultures are often diffusion-limited, leading to oxygen deprivation and necrosis (*McMahon et al., 2012*). There are some overlaps, but also many differences, between gene expression in 3D culture and hypoxic conditions (*DelNero et al., 2015*). These studies emphasize that growth in 3D culture is associated with heterogeneous oxygen concentrations and have found an interdependence between the effects of hypoxia and 3D growth on gene expression. The proteomic changes likely reflect not only oxygen limitation but also other processes connected with 3D growth (e.g., nutrient deprivation, extracellular architecture, and even light penetration). Although the comparisons made here do not address these individual
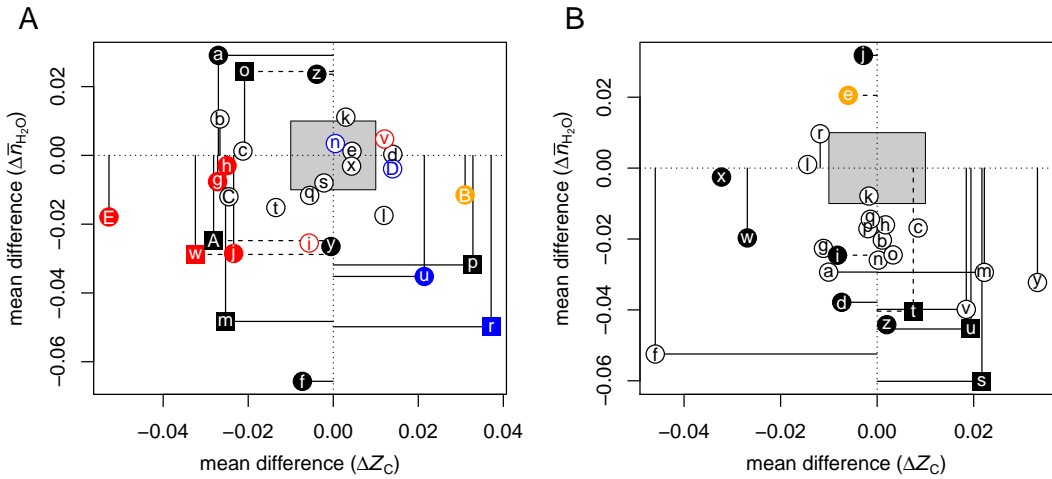

**Figure 2   Compositional analysis of differential protein expression in (A) hypoxia or 3D culture and (B) hyperosmotic stress.** The plots show differences ($\Delta$) between the mean for up-expressed and the mean for down-expressed proteins of average oxidation state of carbon ($Z_C$) and water demand per residue ($\bar{n}_{H_2O}$) for each dataset from Tables 3 and 4. Red, blue, and orange symbols are used to highlight datasets for tumorspheres, reoxygenation or anti-hypoxic treatment, and adipose-derived stem cells, respectively.

factors, they do provide information on whether hypoxia and 3D culture lead to similar changes in the overall chemical composition of proteomes.

Table 3 lists selected proteomic datasets with a minimum of ca. 20 up- and 20 down-expressed proteins in hypoxia or 3D growth. The differences in chemical composition of the differentially expressed proteins are plotted in Fig. 2A. In many experiments, hypoxia or 3D growth induces a proteomic transformation with a significant and/or large decrease of $Z_C$ [ⓐ ⓑ ⓒ ⓖ ⓗ ⓘ ⓜ ⓞ ⓦ Ⓐ Ⓔ]. These datasets cluster around a narrow range of $\Delta Z_C$ ($-0.032$ to $-0.021$), except for dataset Ⓔ (3D growth of colon cancer cells) with much lower $\Delta Z_C$. As extracellular proteins have relatively high $Z_C$ (*Dick, 2014*), the observation in some experiments that hypoxia decreases the abundance of proteins associated with the extracellular matrix (ECM) (*Blankley et al., 2010*) is compatible with the overall expression of more reduced (low-$Z_C$) proteins. Conversely, reoxygenation leads to the formation of more oxidized proteins in the supernatant (-S) and pellet (-P) fractions of isolated chromatin [Ⓡ Ⓤ].

While most studies controlled gas composition to generate hypoxia, two datasets [Ⓒ Ⓓ] are from a study that used cobalt chloride ($CoCl_2$) to induce hypoxia in rat cardiomyocytes; treatment with salidroside (SAL) had anti-hypoxic effects (*Xu et al., 2016*). The $CoCl_2$ and SAL treatments result in the expression of somewhat more reduced and more oxidized proteins, respectively, in agreement with the general trends for hypoxia and reoxygenation experiments.

Two datasets oppose the general trends, showing large and significantly higher $Z_C$ under hypoxia. These datasets were obtained using particular analytical methods or cell types. One of the nonconforming datasets is for the supernatant in a chromatin isolation procedure [Ⓟ], and the other is for adipose-derived stem cells [Ⓑ] (see below).

## Hyperosmotic stress

By hyperosmotic stress is meant a condition that increases the extracellular hypertonicity, or osmolality. The addition of osmolytes (or "cosolvents") lowers the water activity in the medium (*Timasheff, 2002*). Equilibration with hypertonic solutions drives water out of cells, causing cell shrinkage. The selected datasets listed in Table 4 include at least ca. 20 up-expressed and 20 down-expressed proteins in response to high concentrations of NaCl (five studies), glucose (six studies), succinate (one study), KCl (one study), or adaptation to seawater (one study). The proteomic analyses used bacterial, yeast, or mammalian cells, or fish (eel) gills (*Tse et al., 2013*). One study varied temperature along with NaCl concentration (*Kocharunchitt et al., 2012*), and one study reported both transcriptomic and proteomic ratios (*Kohler et al., 2015*).

In the study of *Giardina, Stanley & Chiang (2014)* [ⓞ ⓟ ⓠ], the reported expression ratios for extracellular proteins after transfer from low glucose to high glucose media are nearly all less than 1. Therefore, the "up-expressed" proteins in the comparisons here are taken to be those that have a higher expression ratio than the median in a given experiment. To achieve a sufficient sample size using data from *Chen et al. (2015)* [ⓡ], the comparisons here use a combined set of proteins, i.e., those identified to have the same direction of change in the two treatment conditions (380 and 480 mOsm NaCl) and a significant change in at least one of the conditions.

Figure 2B shows that hyperosmotic stress strongly (CLES $\leq 40\%$) and/or significantly ($p$-value $< 0.05$) induces the formation of proteins with relatively low water demand per residue in 11 datasets [ⓐ ⓑ ⓓ ⓕ ⓘ ⓜ ⓢ ⓣ ⓤ ⓥ ⓩ]. Five of these datasets, including four for bacteria [ⓢ ⓣ ⓤ ⓥ] and one for human cells [ⓜ], also show an increase in $Z_C$. These trends are found in both the transcriptomic [ⓢ ⓣ] and proteomic [ⓤ ⓥ] data from the study of *Kocharunchitt et al. (2012)*.

Four datasets obtained for mammalian cells have low $\Delta Z_C$ with no significant [ⓡ ⓦ ⓧ] or a significantly negative mean difference of $\overline{n}_{H_2O}$ [ⓕ]. Six datasets [ⓗ ⓚ ⓝ ⓞ ⓟ ⓠ] from one study each of yeast and *E. coli*, and of Japanese eels adapted to seawater, have very small mean differences in $Z_C$ and a negative $\Delta\overline{n}_{H_2O}$ that follows the trends of most of the other datasets, but with lower significance ($p$-value $> 0.05$).

The comparisons here show that hyperosmotic stress consistently induces the formation of proteins with lower water demand per residue. In some, but not all, cases, this coincides with an increase in average oxidation state of carbon. Less often, and perhaps specific to mammalian cells, the proteomic composition is shifted toward lower oxidation state of carbon. There are only a couple of datasets, using NaCl treatment [ⓒ ⓘ], that show an increase in water demand per residue.

Notably, two datasets for adipose-derived stem cells oppose the general trends for hypoxic and hyperosmotic conditions (see Fig. 2A [ⓑ] and Fig. 2B [ⓒ]). This intriguing result shows that these stem cells respond to external stresses with proteomic transformations that are chemically similar to those in cancer (Fig. 1).
## Potential diagrams

The correlations of compositional differences (negative $\Delta Z_C$ and $\Delta \bar{n}_{H_2O}$) with hypoxia and hyperosmotic stress can be proposed as resulting from attraction of the proteomes to a context-specific low-energy state. Thermodynamic models can help to illuminate the possible microenvironmental constraints on the observed proteomic transformations. Here, the chemical affinities of stoichiometric formation reactions of proteins were calculated, grouped, and compared in order to estimate the thermodynamic potential for the overall process of proteomic transformation.

The chemical affinity quantifies the potential, or propensity, for a reaction to proceed. It is the infinitesimal change with respect to reaction progress of the negative of the Gibbs energy of the system. The chemical affinity is numerically equal to the "non-standard" or actual (*Warn & Peters, 1996*), "real" (*Zhu & Anderson, 2002*), or "overall" (*Shock, 2009*) negative Gibbs energy of reaction. These energies are not constant, but vary with the chemical potentials, or chemical activities, of species in the reaction. Chemical activity ($a$) and potential ($\mu$) are related through $\mu = \mu° + RT \ln a$, where the standard chemical potentials of particular species ($\mu° = G°$, i.e., standard Gibbs energies) depend only on temperature and pressure.

The equilibrium constant ($K$) for a reaction is given by $\Delta G° = -2.303 RT \log K$, where $\Delta G°$ is the standard Gibbs energy of the reaction, 2.303 stands for the natural logarithm of 10, $R$ is the gas constant, $T$ is temperature in Kelvin, and log denotes the decadic logarithm. The equation used for affinity ($A$) is $A = 2.303 RT \log(K/Q)$, where $Q$ is the activity quotient of the reaction (e.g., *Helgeson, 1979*, Eq. 11.27; *Warn & Peters, 1996*, Eq. 7.14; *Shock, 2009*). Accordingly, the per-residue affinity of Reaction (R1) can be written as

$$A = 2.303 RT (\log K + \bar{n}_{Cys} \log a_{Cys} + \bar{n}_{Glu} \log a_{Glu} + \bar{n}_{Gln} \log a_{Gln}$$
$$+ \bar{n}_{H_2O} \log a_{H_2O} + \bar{n}_{O_2} \log f_{O_2} - \bar{z}_{H^+} pH - \log a_{residue}) \tag{2}$$

where the abbreviations of the amino acids have been substituted for their formulas. Here, $a$ and $f$ stand for chemical activity and fugacity (e.g., $a_{H_2O}$ is water activity, and $f_{O_2}$ is oxygen fugacity). The fugacity, rather than activity, of $O_2$ is used because gaseous oxygen is the reference state most commonly used in previous thermodynamic models. If $a_{O_2}$ were used instead, its values would differ from $f_{O_2}$ according to the solubility of oxygen in water at the given temperature but otherwise the two models would be thermodynamically equivalent. The overbar notation ($\bar{n}$ and $\bar{z}$) signifies that the coefficients in Reaction (R1) are each divided by the length (number of amino acids) of the protein sequence. Likewise, the elemental composition and standard Gibbs energy per residue are those of the ionized protein (with formula $C_c H_{h+z} N_n O_o S_s^z$) divided by the length of the protein.

The standard Gibbs energies of species at 37 °C and 1 bar were calculated with **CHNOSZ** (*Dick, 2008*) using equations and data taken from *Wagman et al. (1982)* and *Kelley (1960)* ($O_{2(g)}$), *Johnson, Oelkers & Helgeson (1992)* and references therein ($H_2O$), and using the Helgeson–Kirkham–Flowers equations of state (*Helgeson, Kirkham & Flowers, 1981*) with data taken from *Amend & Helgeson (1997)* and *Dick, LaRowe & Helgeson (2006)* (amino acids), and from *Dick, LaRowe & Helgeson (2006)* and *LaRowe & Dick (2012)* (amino acid group additivity for proteins).

In previous calculations, activities of the amino acid basis species and protein residues were set to $10^{-4}$ and $10^0$, respectively (*Dick, 2016*). As long as constant total activity of residues is assumed, the specific value does not greatly affect the outcome of the calculations; here it is kept at $10^0$. Revised activities of the amino acid basis species, corresponding to mean concentrations in human plasma (*Tcherkas & Denisenko, 2001*), are used here: $10^{-3.6}$ (cysteine), $10^{-4.5}$ (glutamic acid) and $10^{-3.2}$ (glutamine). Adopting these activities of basis species, instead of $10^{-4}$, lowers the calculated equipotential lines for proteomic transformations by about 0.5 to 1 $\log a_{H_2O}$ (see below). Accounting for protein ionization, with pH set to 7, also lowers the equipotential lines, by about 1 $\log a_{H_2O}$ compared to calculations for nonionized proteins.

It follows from Eq. (2) that varying the fugacity of $O_2$ and activity of $H_2O$ alters the chemical affinity for formation of proteins by a specific amount depending on their chemical composition. For example, Figure 5A of *Dick (2016)* shows that decreasing $\log f_{O_2}$ is relatively more favorable for the formation of up-expressed than down-expressed proteins in a particular cancer dataset (*Knol et al., 2014*; ⓦ in Table 1). This tendency is consistent with the lower $Z_C$ of these up-expressed proteins, which is unlike most other datasets for CRC (Fig. 1A).

How can the affinities of groups, rather than individual proteins, be compared? One method is based on differences in the ranks of chemical affinities of proteins between groups (*Dick, 2016*). Using this method, the affinities of all of the proteins in a dataset are ranked; the ranks are then summed for proteins in the up- and down-expressed groups ($r_{up}$ and $r_{down}$). Before taking the difference, the ranks are multiplied by a weighting factor to account for the different numbers of proteins in the groups ($n = n_{up} + n_{down}$). This weighted rank difference (WRD) of affinity summarizes the estimates of the differential potential for formation:

$$\text{WRD} = 2\left(\frac{n_{down}}{n}\sum r_{up} - \frac{n_{up}}{n}\sum r_{down}\right). \tag{3}$$

On a contour diagram of the WRD of affinity (referred to here as a "potential diagram"), the line of zero WRD represents a rank-wise equal affinity (or "equipotential line") for formation of proteins in the two groups.

To characterize the general trends, diagrams were made for groups of proteomic datasets with similar compositional features. For pancreatic cancer, there are 11 datasets with $\Delta Z_C > 0.01$ (i.e., to the right of the gray box in Fig. 1B) and for which the mean difference of $\overline{n}_{H_2O}$ is neither significant (low $p$-value) nor large (high CLES). Conversely, there are 8 datasets for pancreatic cancer with $\Delta \overline{n}_{H_2O} > 0.01$ and for which the mean difference of $Z_C$ is neither large nor significant. Similarly, weighted rank-difference diagrams were constructed for 13 ($\Delta Z_C > 0.01$) and 10 ($\Delta \overline{n}_{H_2O} > 0.01$) datasets for CRC, 8 datasets for hypoxia ($\Delta Z_C < -0.01$), and 12 datasets for hyperosmotic stress ($\Delta \overline{n}_{H_2O} < -0.01$). The individual diagrams for each of these groups are presented in Fig. S3.

In order to observe the central tendencies among the various datasets, the potential diagrams for each group in Fig. S3 were combined by taking the arithmetic mean of the

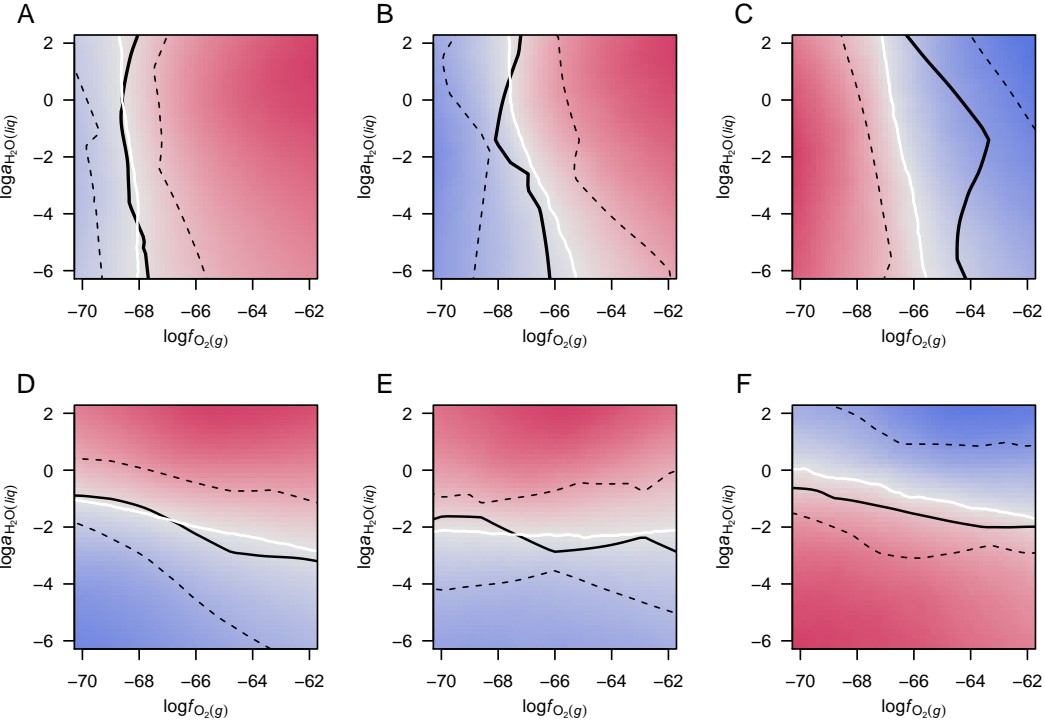

**Figure 3  Merged potential diagrams for proteomic transformations.** Plots are shown for (A) 13 datasets for colorectal cancer and (B) 11 datasets for pancreatic cancer with $\Delta Z_C > 0.01$, (C) eight datasets for hypoxia or 3D culture with $\Delta Z_C < -0.01$, (D) 10 datasets for colorectal cancer and (E) eight datasets for pancreatic cancer with $\Delta \bar{n}_{H_2O} > 0.01$, and (F) 12 datasets for hyperosmotic stress with $\Delta \bar{n}_{H_2O} < -0.01$. Red and blue colors denote higher relative potential for formation of up- and down-expressed proteins, respectively. White lines are equipotential lines, where the mean weighted rank difference of affinity (WRD; Eq. (3)) of the included datasets is 0; black lines show the median and interquartile range of the WRD = 0 lines for individual datasets (Fig. S3). See text for details.

WRD at all grid points in $\log f_{O_2}$–$\log a_{H_2O}$ space. The resulting diagrams (Fig. 3) have equipotential lines, shown in white, and zones of positive and negative WRD of affinity, i.e., greater relative potential for formation of up- and down-expressed groups of proteins, colored red and blue, respectively.

The solid black lines in Fig. 3 show the median position along the *x*- or *y*-axis for the equipotential lines in each group (Fig. S3), and the dashed black lines are positioned at the 1st and 3rd quartiles. The interquartile ranges for the cancer groups are smaller than those for hypoxia, but less so for hyperosmotic stress. The smaller range would be expected if the cancer datasets reflected a somewhat narrower set of conditions than the datasets for experiments with hypoxia; the latter represent a wide variety of organisms, cell types, and laboratory conditions (Table 3).

## DISCUSSION

Calculations of the average oxidation state of carbon and water demand per residue, derived from elemental stoichiometry, provide information on the microenvironmental factors affecting differential protein expression in cancer and laboratory experiments. Hypoxia

or hyperosmotic stress generally induces the expression of proteins with lower overall oxidation state of carbon or lower water demand per residue, respectively, compared to down-expressed proteins. In contrast, proteomes of CRC and pancreatic cancer are often characterized by greater water demand per residue or oxidation state of carbon. The formation of more highly oxidized proteins despite the hypoxic conditions of many tumors hints at a complex set of microenvironmental–cellular interactions in cancer.

Plots of data from experiments with hypoxia and hyperosmotic stress illuminate two dimensions of possible compositional attraction to a low-energy state (Fig. 2). A thermodynamic model quantifies the altered potential for proteomic transformation in response to changing oxygen fugacity and water activity. The equipotential lines for cancer proteomes with high differential water demand lie between $\log a_{H_2O} = -1$ to $-3$, while the potential threshold for transformation of proteomes in hyperosmotic stress is closer to unit activity of water ($\log a_{H_2O} = -0$ to $-2$) (Figs. 3D–3F). Although there is considerable variability among the individual datasets (Fig. S3), the merged diagrams demonstrate a physiologically realistic range for the activity of water. Water activity in cells is close to one, but restricted diffusion of $H_2O$ in "osmotically inactive" regions of cells (Model, 2014) could result in locally lower water activities. The present findings provide evidence that the molecular processes regulating proteomic transformations operate within the chemical constraints of subcellular regions of depleted water activity.

The finding of a frequently positive water demand for the transformation between normal and cancer proteomes offers a new perspective on the biochemistry of hydration in cancer. The thermodynamic calculations predict that, in contrast to hyperosmotic stress, proteomes of cancer tissues are stabilized by increasing water activity. A higher than normal water activity would be consistent with the greater hydration of tissue that is apparent in spectroscopic analysis of breast cancer tissue (e.g., Abramczyk et al., 2014). Speculatively, the relatively high water content needed for embryonic development (Moulton, 1923) could be recreated in cancer cells if they revert to an embryonic mode of growth (McIntyre, 2006).

The equipotentials for transformation of proteomes in cancer cluster near an oxygen fugacity of ca. $10^{-68}$ to $10^{-66}$. The oxygen fugacity should be interpreted not as actual oxygen concentration, rather as a internal scale of oxidation potential. Oxygen fugacity and water activity can be converted to the Eh scale for redox potential, giving values that are comparable to other biochemical measurements (Dick, 2016).

Although cancer proteomes are obtained from tissues that are likely derived from hypoxic tumor environments, their differential expression is most often in favor of oxidized proteins (Figs. 1A and 1B). What are some explanations for this finding? Perhaps the relatively high $\log f_{O_2}$ threshold for chemical transformation of hypoxia-responsive proteins could support a buffering action that potentiates the formation of relatively oxidized proteins in cancer (compare the median and quartiles in Fig. 3C with those in Figs. 3A and 3B). This speculative hypothesis requires a division of the cellular proteome into localized, chemically interacting subsystems. Alternatively, the development of a high oxidation potential in cancer cells may be associated with a higher concentration of mitochondrially produced reactive oxygen species (ROS). Neither of these possibilities

addresses the magnitude of the chemical differences in the proteomes, and the question remains: where do the electrons go?

A plausible hypothesis comes from considering the different oxidation states of biomolecules. Fatty acids are reduced compared to amino acids, nucleotides, and saccharides (*Amend et al., 2013*). In parallel with the formation of more reduced proteins, hypoxia induces the accumulation of lipids in cell culture (*Gordon, Barcza & Bush, 1977*). Cancer cells are also known for increased lipid synthesis. Lipid droplets, which are derived from the endoplasmic reticulum (ER), form in great quantities in cancer cells (*Koizume & Miyagi, 2016*). Assuming that lipids are synthesized from relatively oxidized metabolic precursors, their formation requires a source of electrons. These considerations lead to the hypothesis that increased lipid synthesis is coupled to the oxidation of the proteome.

Calculations that combine proteomic and cellular data can be used to quantify a hypothetical redox balance between cellular lipids and proteins. The major assumptions in the calculations here are that the overall cellular oxidation state of carbon is the same in cancer and hypoxia, and that changes in this cellular oxidation state are brought about by altering only the numbers of lipid and protein molecules. The overall chemical composition of the lipids is assumed to be constant, but the proteins are assigned different values of $Z_C$. These simplifying assumptions are meant to pose quantifiable "what if" questions, to serve as points of reference about the range of molecular composition of cells (*Milo & Phillips, 2015*).

The worked-out calculation is shown in Fig. 4. The lipid:protein ratio in hypoxia is taken from *Gordon, Barcza & Bush (1977)*, and ballpark values for the differences in $Z_C$ of proteins in hypoxia and cancer are from the present study. Notably, the lipid:protein weight ratio in hypoxia (0.19) is higher than in normal cells (i.e., 0.15 using data from *Gordon, Barcza & Bush, 1977* or 0.16 using data compiled by *Milo & Phillips, 2015* for *E. coli*). The calculation indicates that an increase of the lipid:protein weight ratio in cancer cells by ca. 20% over that in hypoxic normal cells could provide an electron sink that is large enough to take up the electrons released by oxidation of the proteome in hypoxic normal cells to generate that in hypoxic cancer cells. That proteomic transformation is quantified here by an increase of $\Delta Z_C$ from ca. $-0.03$ to $0.03$, both relative to non-hypoxic normal cells (Fig. 4).

As found by Raman spectroscopy, levels of both lipids and proteins are elevated in colorectal cancer (*Stone et al., 2004*). Lipid droplets are formed extensively in CRC stem cells (*Tirinato et al., 2015*), suggestive of a higher lipid:protein ratio than either cancer or normal epithelial cells. In contrast to CRC, lipids are decreased in breast cancer compared to normal breast tissue (*Frank, McCreery & Redd, 1995*; *Stone et al., 2004*). Given a lower lipid content, and therefore smaller electron sink, one might expect that proteomes in breast cancer are oxidized to a lesser extent than those in CRC and pancreatic cancer. Other factors that affect the systemic redox balance, such as a more reduced gut microbiome in CRC (*Dick, 2016*) and metabolic coupling between epithelial and stromal cells, may be important for an accurate account of the compositional relationships among biomacromolecules.

These compositional and thermodynamic analyses support the notion that changes in bulk chemical composition of cells and the microenvironment have a significant role

| R code | comments / results (rounded) |
|---|---|
| ```library(CHNOSZ); data(thermo)``` <br> ```lipid <- "C55H102O6"``` <br> ```ZC_lipid <- ZC(lipid)``` | load **CHNOSZ** package <br> *value*: lipid formula [1] and $Z_C$ <br> **-1.64** |
| ```library(canprot); data(canprot)``` <br> ```protein <- colMeans(protein.formula(human_base))``` <br><br> ```ZC_protein <- ZC(protein)``` | load **canprot** package <br> *value*: initial protein formula [2] and $Z_C$ <br> $C_{2683}H_{4236}N_{749}O_{810}S_{24}$ <br> **-0.12** |
| ```ΔZC_hypoxia <- -0.03``` <br> ```ΔZC_cancer <- 0.03``` | *values*: $\Delta Z_C$ of proteins [3] |
| ```LP_hypoxia <- (114.7 + 16.4 + 41.9 + 18.6) / 1000``` | *value*: L:P weight ratio in hypoxia [4] <br> **0.19** |
| ```LP_mass2mol <- (makeup(lipid)["C"] / mass(lipid)) /``` <br> ```  (makeup(protein)["C"] / mass(protein))``` <br> ```LPC_hypoxia <- LP_hypoxia * LP_mass2mol``` | *conversion*: L:P C-atom ratio in hypoxia <br><br> **0.28** |
| ```ZC_cell <- function(LPC, ΔZC)``` <br> ``` ZC_lipid * LPC/(LPC+1) +``` <br> ``` (ZC_protein + ΔZC) * 1/(LPC+1)``` <br> ```ZC_hypoxia <- ZC_cell(LPC_hypoxia, ΔZC_hypoxia)``` | *assumption*: cell $\Delta Z_C$ is sum of L and P contributions <br><br> *calculation*: cell $Z_C$ in hypoxia <br> **-0.47** |
| ```ZC_cancer <- function(LPC_cancer)``` <br> ``` ZC_cell(LPC_cancer, ΔZC_cancer)``` | *unknown*: L:P C-atom ratio in cancer |
| ```ΔZC <- function(LPC_cancer)``` <br> ``` ZC_cancer(LPC_cancer) - ZC_hypoxia``` <br> ```LPC_cancer <- uniroot(ΔZC, c(0, 1))$root``` | *assumption*: hypoxia and cancer have same cell $Z_C$ <br> *calculation*: L:P C-atom ratio in cancer <br> **0.33** |
| ```LP_cancer <- LPC_cancer / LP_mass2mol``` | *conversion*: L:P weight ratio in cancer <br> **0.23** |
| ```LP_diff <- 100*(LP_cancer - LP_hypoxia) / LP_hypoxia``` | *comparison*: percent difference between hypoxia and cancer <br> **19** |

**Figure 4** **A computer-aided "back of the envelope" calculation to estimate the lipid to protein ratio (L:P) in cancer cells and the percent difference from normal cells in hypoxic conditions.** Bold text indicates function definitions (R code) or numerical results (comments/results (rounded)). Numerical values are taken from [1] the chemical formula of 1-palmitoyl-2,3-dioleoyl-glycerol, given as an example of a triacylglycerol (triglyceride) in the chapter on lipid metabolism in *Voet, Voet & Pratt (2013)*, [2] the average chemical formula of proteins in the UniProt human proteome, for which amino acid compositions are stored in `human_base.Rdata` in the **canprot** package, [3] this study, and [4] Table 2 of *Gordon, Barcza & Bush (1977)* (mouse cells grown in hypoxic conditions).

in shaping the differential expression of proteins. The analysis done here is primarily concerned with top-down causal factors (physical constraints on protein synthesis and degradation), but does not preclude a major role for bottom-up factors (e.g., regulation of gene expression). Speculatively, further applications of these methods could be used to predict the ability of chemotherapy or other treatments to reduce or reverse the potential for formation of the proteins required by cancer cells. Based on the current findings, a decreased proteomic oxidation and/or hydration state may emerge as one aspect of beneficial treatments.

This approach to the data differs from conventional interpretations of proteomic data that are based on the functions of proteins. Nevertheless, the scope of explanations dealing with functions and molecular interactions offers limited insight on the high-level organization of proteomes in a cellular and microenvironmental context. Although a variety of bioinformatics tools are available for functional interpretations

(*Laukens, Naulaerts & Berghe, 2015*), none so far addresses the overall chemical requirements of proteomic transformations. The compositional and thermodynamic descriptions presented here encourage a fresh look at the question, "What is cancer made of?"

## CONCLUSION

Although many hypoxia experiments induce the formation of proteins with lower oxidation state of carbon ($Z_C$), the up-expressed proteins in colorectal and pancreatic cancer are often relatively oxidized compared to the down-expressed ones. Hyperosmotic stress in the laboratory leads to the formation of proteins with relatively low water demand per residue ($\overline{n}_{H_2O}$), but cancer proteomes often show the opposite trend, with up-expressed proteins having higher average $\overline{n}_{H_2O}$ than down-expressed ones.

The global proteomic differences can be described as compositional changes in terms of chemical basis species and quantified in a thermodynamic framework. A positive thermodynamic potential for each proteomic transformation is predicted in a specific range of oxidation and hydration potential. However, the distribution of biomolecules other than proteins should also be considered to account for changes in cellular redox balance. An electron sink associated with a ca. 20% greater lipid to protein ratio in cancer compared to normal hypoxic cells would be sufficient to balance the electrons released by the formation of more oxidized proteins in CRC and pancreatic cancer. It thus appears possible that a redox disproportionation develops in some cancers, leading to pools of both more reduced and more oxidized macromolecules compared to normal conditions.

## ACKNOWLEDGEMENTS

I thank Apar Prasad and Jack Staunton for their comments and suggestions, and Ming-Chih Lai for providing data generated in the study of *Lai, Chang & Sun (2016)* and for giving permission to include it here.

### Funding

The author received no funding for this work.

### Competing Interests

The author declares there are no competing interests.

### Author Contributions

- Jeffrey M. Dick analyzed the data and wrote the paper.

### Supplemental Information

Supplemental information for this article can be found online at http://dx.doi.org/10.7717/peerj.3421#supplemental-information.

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
