# Peer review of "Chemical composition and the potential for proteomic transformation in cancer, hypoxia, and hyperosmotic stress"

_PeerJ, doi:10.7717/peerj.3421_

## Round 0.1 · original submission · Minor Revisions

I suggest not to consider the experimental data (to test the outcome of the oxidative stress predictions on 3D cultures by mass spectrometry) requested from one of the reviewers since they are not pertinent.

Reviewer 1 ·

Basic reporting

The abstract is concise and descriptive.
The document presents several typos (see below). I suggest having a native English speaker revising the grammar and the sentence flow throughout the manuscript.

Examples:
Line 4: myriad (of) molecular-level
Lines 34, 41: incorrect use of the word ‘for’
Line 46: ‘the’ is missing – ‘the complexity of (the) underlying…’
Line 56: ‘is inherent to the’ and not ‘is inherent in the’
Line 123: Consider change ‘must take account of the gain’ to ‘must consider the gain’


The author stated that there are few studies available addressing the thermodynamic potential related to proteome changes, though none is cited. Available literature should be discussed and cited properly.
Tables 1-4 are well structured and their captions are adequate. Still, the meaning of n1 and n2 in the table legend is lacking.

A good explanation for the thermodynamic concepts of average state of carbon and of water demand per residue is provided. Still, I suggest to introduce the commonly used term of ‘hydration shell’ to discern ‘protein hydration’ from ‘hydration state’.

Equation R1 only accounts for Cys, Glu and Gln from the 20 amino acids (aa) that compose proteins. While this is a mathematical model, a brief comment is expected regarding the choice of these aa or the exclusion of the remaining 17 aa.

The mathematical model was applied to understand the thermodynamic basis of the proteomic changes in several pathological and experimental conditions such as colorectal cancer, pancreatic cancer, hypoxia and hyperosmotic stress. Even though these are provided as working examples, some references are needed. For instance, line 157 (colorectal cancer) and line 222 (hypoxia) require referencing.

Lines 168-169: Are you sure that n1 and n2 are always the number of down- and up-regulated proteins in cancer versus normal tissue? For instance, sets b, c, and p from Table 1 compare carcinoma samples with adenoma samples. Can adenoma be regarded as a normal tissue? Shouldn’t it be classified as a benign tumor? Please revise and rephrase it accordingly.

Experimental design

The goal is well presented and the paper addresses a relevant, yet obscure, question that may have impact in the (bio)medical/health sciences.

Line92: Name the other sources.

I am not sure about the inclusion of samples of chronic pancreatitis, autoimmune pancreatitis and diabetes mellitus (set d, l and q from Table 2) in the analysis due to the different pathological backgrounds. At least, they should be regarded as potential outliers and not just set s and, thus, they should be discussed accordingly in the R&D section.
Also, regarding ‘hypoxia and 3D culture’, I do not feel that cells cultured as spheroids or other 3D models exactly replicate hypoxia models per se. While oxygen tension can be small in the core of the organoid/spheroid, there are other confounders that can explain changes in ZC and water demand per residue, such as nutrient deprival (and thus oxidation of unusual substrates), different extracellular architectures and even light penetration (which may affect redox reactions).

Validity of the findings

My concern is with data heterogeneity. For instance, for pancreatic cancer, inclusion of diabetes mellitus, chronic pancreatitis and autoimmune pancreatitis may have biased diagrams from Figure 3. Additionally, inclusion of 3D cell culture models to mimic hypoxia is somehow controversial. Thus, I suggest to exclude these data from the thermodynamic analysis or discuss it in Supplementary data.

Additional comments

The author provides an interesting topic on thermodynamic variables driving the proteomic changes in a variety of pathological and experimental conditions. If the concerns exposed above are duly addressed or justified I support the publication.

Reviewer 2 ·

Basic reporting

no comment

Experimental design

The paper is substantially a good mathematical work based on deposited biological data, but in my opinion there is not a perfect match with the aim and scope of the journal that does not publish Mathematical Sciences paper.

Validity of the findings

The reported results may represent a new point of view of cellular and tissues behaviour dynamics, but they are based only on literature datasets without any direct examination of the formulated hypothesis, with a consequential speculation bias.

Additional comments

I suggest the authors to improve the paper with some experimental data performed by themselves. For example they could test the outcome of the oxidative stress predictions on 3D cultures by mass spectrometry.

·

Basic reporting

The paper is clearly written. I commend the author for sharing the raw data in an R package.

1. I recommend to shorten the abstract.

2. Considering ‘self-contained’ policy of the journal, I think it may not be appropriate to mention preliminary analysis of breast cancer, for which no data is shown, even if it is in the discussion section.

Figures are appropriately labeled.
3. Perhaps Figure 4 could be moved to supplementary information.

4. Figure 3 has to be placed after equation 3.

5. I also think subfigures in Figure 3 could be arranged in a regular grid.

Experimental design

1. Is total protein charge Z missing from the equation 1? It is part of the equation in Dick, 2014.

2. It may be mentioned in Dick, 2016 but I think it is necessary to explain why QEC amino acids are sufficient for assessing Zc of the protein.

3. Is there any known functional bias (GO enrichment) in QEC+ biased proteins?

4. A Figure connecting Figure 1 and Figure 2 and drawing parallels between cancer cells and hypoxic/hyperosmotic cells could be helpful.

5. Although it is mentioned in the caption, it is not that clear how to interpret the sign of mean differences, if it is up or down regulated proteins that have increase of Zc and NH2O.

6. It is not explained in the text why oxygen fugacity is added to equation 2.

Validity of the findings

1. My major concern is disregarding the effect of metabolome changes on Zc and N_H2O since metabolic changes is one of the hallmarks of cancer.

2. I think there is a discussion point missing on whether the changes in protein concentrations are the result of physical constraints on protein synthesis/degradation or a manifestation of cellular response in terms of gene expression levels.

3. Speculative statements in conclusions should be moved to Discussion section.

---

## Round 0.2 · accepted · Accept

I really appreciated how the authors performed the corrections suggested and discussed the points surfaced by the reviewers.